# Experience of Chinese Recent Retirees on the Effects of Retirement on Healthy Ageing in Shenzhen and Hong Kong

**DOI:** 10.3390/ijerph20042820

**Published:** 2023-02-05

**Authors:** Daniel W. L. Lai, Yong-Xin Ruan, Julia Juan Wang, Emma H. S. Liu, Jia-Jia Zhou

**Affiliations:** 1Faculty of Social Sciences, Hong Kong Baptist University, Hong Kong, China; 2Department of Social Work, The Chinese University of Hong Kong, Hong Kong, China; 3Department of Elderly Healthcare, Shenzhen Polytechnic College, Shenzhen 518055, China; 4Department of Applied Social Sciences, The Hong Kong Polytechnic University, Hong Kong, China

**Keywords:** retirement, healthy ageing, participation, financial benefits, China, Hong Kong

## Abstract

This study examined perspectives of recent retirees in Shenzhen and Hong Kong on how retirement influenced their healthy ageing. It investigated retirees’ perceptions of healthy ageing and the ways in which healthy ageing connected with retirees’ transition into retirement. A qualitative design with narrative interviews was used to interview twelve recent retirees in Shenzhen and thirteen in Hong Kong. The participants elaborated their perspectives on healthy ageing, which covered physical, mental, social, and financial domains. Retirees in both cities identified healthy ageing as maintaining an independent life and avoiding becoming a burden on family members. This study found that retirement declined physical health (in parallel with raised awareness of health promotion), posed both negative and positive influences on mental health, and shrank peripheral social networks of retirees. In addition, regional social welfare systems have different impacts on retirees’ financial security and social participation. Retirees in Hong Kong reported higher stress of financial security and a strong desire for labor participation. Migrant–local welfare gaps were documented by retirees in Shenzhen. This study suggested that retirement planning, establishing a multi-pillar retirement-protection system, and narrowing the welfare gap between migrants and local residents should be implemented to reinforce healthy ageing.

## 1. Background

According to the World Health Organization (WHO), healthy ageing is “the process of developing and maintaining the functional ability that enables well-being in older age” [1]. Specifically speaking, healthy ageing is a multidimensional concept. A review study has shown that previous studies mainly measured healthy ageing in physical, mental, and social domains, with limited research focused on other components (e.g., financial security) [2]. There have been mixed results regarding the impact of retirement on healthy ageing. Some older people tend to report poorer health due to a sense of uselessness after retirement [3]. Meanwhile, with an increase in healthy behaviors without physical and psychological pressures from work, some older people tend to report better health [4]. Without work pressure, retirement can also promote retirees’ well-being and reduce their distress and depression [5]. Retirees tend to have a larger size of familial and friend social networks since they have more time to interact with family members and friends [6]. Several scholars also identified a positive relationship between retirement and participation in social activities [6,7].

Despite the increasing number of older people in Mainland China and Hong Kong [8,9], a comprehensive retirement-protection system has not yet been well prepared in both regions [10,11]. For instance, policies in Hong Kong relating to retirement age and pension schemes have remained nearly unchanged for half a century, greatly deteriorating the health status and financial security of retirees [12]. It is within this context that it is significant to explore how retirement influences healthy ageing in Mainland China and Hong Kong so as to inform relevant policies and improvement in services.

In this study, the influences of retirement on healthy ageing were explored, from the perspectives of recent retirees in Hong Kong and Shenzhen. Taking into consideration the diverse socio-economic contexts of retirement, Chinese retirees from these two cities were selected to enhance understanding of the influences of retirement, an important life course change, on healthy ageing, which serves as an important life course goal.

## 2. Literature Review

### 2.1. Influence of Retirement on Healthy Ageing

When older people experience the transition to retirement, they go through changes in their physical health, mental health, and social well-being. In terms of physical health, there were mixed results regarding the relationship between retirement and physical health in Mainland China. Some researchers found that retirement reduced male retirees’ health in China [13], while other researchers did not detect any significant effects of retirement on individual physical health [14]. On the other hand, another study reported a positive association between retirement and self-rated physical health [15]. 

For mental health, many studies reported that retirement had a positive influence on the mental health of older people. Owing to the relief from work pressure, a systematic review has indicated that retirement can promote retirees’ well-being and reduce their distress and depression [5]. This conclusion was identified in both Mainland China and Hong Kong where older people reported decreased depression symptoms and improved psychological well-being after retirement [15,16]. On the other hand, a recent study in Canada has illustrated that going through retirement can be an identity disruption for many older adults, posing risks to their mental well-being [17]. Research findings also indicated that retirement reduced stress in Chinese men but had the contrary impact on women [18]. 

In terms of social well-being, there were limited studies investigating the impact of retirement on social networks and the results are mixed. On one hand, retirees tend to have a larger size of familial and friend social networks since they have more time to interact with family members and friends, or participate in social activities [6,7]. Kalbarczyk and Łopaciuk-Gonczaryk [19] revealed that different types of post-retirement activities were complementary with each other, where volunteering was most attractive for most active groups of older people [19]. In contrast with active social engagement, Korean retirees showed a progressive decrease in the frequency of visiting friends and an abrupt reduction in social gatherings [20]. Some scholars believed that retirement has limited effects on social networks [21] and a small change in activity participation after retirement [19]. Nevertheless, research on the effects of retirement on social well-being in Chinese context has been scant. 

There were several research limitations in the existing literature on the effects of retirement on healthy ageing. First, there were conflicting findings regarding physical, mental, and social domains of healthy ageing after retirement among older adults. Second, few studies examined the association between retirement and financial security. In view of considering healthy ageing as a multi-dimensional conception, studies should cover a full picture of multiple domains of healthy ageing. Third, some quantitative studies investigated the influence of retirement on health under the voluntary retirement system. Mainland China generally implements a mandatory retirement system, which may trigger different influences of retirement on healthy ageing. Therefore, to address these limitations, there is a need for further studies of the effects of retirement on the multiple domains of healthy ageing in the Chinese context.

### 2.2. Retirement Contexts in Shenzhen and Hong Kong

Shenzhen and Hong Kong are neighboring cities located within the Pearl River Delta in China. Both cities are densely populated. The total population of Hong Kong was 7,486,400 in 2018. Of these, 1,870,000 people were aged over 60 [22]. The total population with permanent residence in Shenzhen was 12,528,300 in 2017 [23]. The number of people aged over 60 with permanent residence in Shenzhen was 1,100,000 [24]. In other words, both cities have a gradually ageing population, and they face challenges in managing their ageing populations.

However, there are differences between Shenzhen and Hong Kong from other perspectives including the retirement system, cultural influence, and social welfare system. Hong Kong has a semi-voluntary retirement system as there is no standard mandatory retirement age for non-government institutions or companies, but this is not the case for governmental departments [12,25]. On the contrary, Mainland China has a mandatory retirement age regardless of workers’ occupations [26]. 

Both cities have been influenced by traditional Chinese values. Nevertheless, Hong Kong, as a former British colony and currently a Special Administrative Region of China under “One Country Two Systems”, has integrated more western culture into its local culture [27]. In contrast, Shenzhen is a Chinese city directly under the governance of the central Chinese government, and is a city full of migrants from other cities in China. Therefore, the culture in Shenzhen has largely been influenced by social norms or lifestyles from other non-Cantonese parts of China [28]. 

In terms of welfare provision, both cities have different types of social welfare systems. In Hong Kong, the Mandatory Provident Fund (MPF) System is one of the most important retirement-protection schemes. It is a defined-contribution scheme that is employment based, privately managed, and mandatory. Retirees over 60 years old can receive benefits in a lump sum or in installments [29]. The Hong Kong government has also introduced several types of subsidies and benefits for permanent senior residents over 65 years old, such as an old-age allowance, a senior-citizen card scheme, and health-care vouchers. Older people in Shenzhen can enjoy state-led pension insurance and medical insurance if their accumulated contribution reaches 15 years [30]. Senior citizens over 60 years old are also entitled to subsidies and benefits, including an old-age allowance, free physical examinations, and community-care vouchers, but some of those benefits require household registration [31,32,33]. 

### 2.3. Conceptual Framework

Based on the WHO definition of healthy ageing, the term “functional ability” includes individual intrinsic capacities, environmental contexts (e.g., social welfare), and interactions (e.g., participation in community) [1]. In accordance with the categorizations of domains of healthy ageing discussed above, a conceptual framework was developed (Figure 1). The conceptual framework served to facilitate our understanding of older people’s experiences of their post-retirement life and guided the analysis of this study. With respect to the transition of retirement, financial protection and social policy were two primary factors influencing the post-retirement life of older adults. For each domain of healthy ageing, the effects of financial and policy factors related to retirement will be analyzed. For example, retirement benefits shaped the context that brings financial security to older people. The social domain of healthy ageing could be affected by employment situations in the labor market and community programs and services. The effects of retirement on multiple domains of healthy ageing will be presented successively.

## 3. Research Methodology

### 3.1. Research Objectives and Questions

The aim of this study is to examine how the retirement of older Chinese in different socio-cultural contexts differs in terms of its implications on their health as they age after retirement. Two key research questions were addressed:

What are the perspectives toward healthy ageing of the older Chinese in Shenzhen and Hong Kong?How does retirement influence the healthy ageing of the older Chinese in Shenzhen and Hong Kong?

### 3.2. Research Design

A narrative interview was used in this study, which is a research method to collect people’s stories [34]. It also has been applied by researchers to explore the experience of older people toward health [35]. Since this study aims to investigate the perceptions of recent retirees toward healthy ageing and retirement, and the information could be revealed through participants’ stories, the narrative interview is a suitable method for this study. Following the process of conducting narrative interview outlined by Muylaert et al. [36], researchers generated “exmanent questions” (topic guides developed from the researcher’s approach to the research focus) to lead the narration, but these gradually transformed into “immanent questions” (themes raised by participants). Participants were the active characters leading the content and pace of interviews [34]. As retirement is a significant event in the life course of older people, narrative interviews could contribute to obtaining a closer observation of the timeline context and considerations behind retirees’ experiences. 

### 3.3. Participants and Data Collection

Older Chinese residing in Shenzhen and Hong Kong who are retired are the target participants for this research. Since the normal retirement age in Shenzhen and Hong Kong varies according to one’s type of employment and business nature of the employers, this study purposely included retirees aged 50 or older and those who had retired within five years at the time of data collection. Efforts were made in recruitment to find participants from different gender groups, age cohorts, and education/employment backgrounds. A snowball sampling strategy was used in the recruitment of participants. Participants were first recruited through distributing recruitment information in the social service agencies serving older people in Shenzhen and Hong Kong. Then the initially recruited participants recommended acquaintances based on the inclusion criteria and their willingness to participate in the research. Data collection took place between September 2018 and May 2019.

Data were collected through face-to-face in-depth interviews by the second and third authors, who have degrees in social work disciplines and possess experience in conducting qualitative interviews. The interviews were either conducted in Putonghua or Cantonese, depending upon the choice of the participants. Each interview lasted for approximately 90 min. Interviews were conducted in places that participants felt comfortable such as community centers or public meeting venues with an appropriate level of privacy and little noise interference. Pilot tests were conducted on a few recent retirees in Shenzhen and Hong Kong to verify the relevancy and feasibility of interview questions. Unstructured questions were asked to encourage participants to share their stories, including “What are your employment trajectories (pathways and experiences)?”, “What does healthy ageing mean to you?”, and “How does retirement influence your healthy ageing?”. Either written or verbal consent was obtained from the participants before the interviews. 

As a result, a total of twenty-five interviews were conducted. Twelve interviews were conducted in Shenzhen (five men and seven women), while thirteen interviews (six men and seven women) were conducted in Hong Kong. The participants were 50 to 65 years old, and they had been retired for 3 years on average. The demographics of these two groups of participants are presented in Table 1.

### 3.4. Data Analysis

All interviews were audio recorded, and field notes were taken to establish the dependability of this research. The second author transcribed the interviews into written simplified Chinese transcripts and analyzed them using thematic analysis following the guidelines of Braun and Clarke [37]. Firstly, the team members familiarized themselves with the transcripts and field notes. Then, the team worked together to develop a coding frame based on the research questions and the conceptual framework that this study was based upon, which included definitions and example of codes. To validate the coding frame, two team members were involved in coding the transcripts separately and occasionally discussed inconsistencies. After all the transcripts were coded, these team members further discussed connections between the codes and associated identified themes. During this process, discussion of the revision of the identified codes and thoughts on organizing themes were documented so that other team members could review and supervise the process. It was not until consensus was reached on the organization of the themes and ensuring that they represented the participants’ stories that the final themes were determined.

During the research process, credibility, transferability, dependability, and confirmability were achieved to ensure the trustworthiness of the thematic analysis [38]. Credibility was achieved through using member checking. In this study, each of the participants, though only interviewed once, was asked to re-confirm and verify the interpretations during the interview as well as through verifying findings and interpretation obtained in previously completed interviews. To achieve transferability, thick descriptions were provided in this study by showing quotes supporting the specific themes or results presented. Dependability refers to the process of research being “logical, traceable, and clearly documented” [38] (p. 3). Thus, in our study, different research team members were involved in auditing these requirements during the data collection and analysis process. For example, interviews conducted by one interviewer were objectively reviewed by another researcher on the team. Coding and theme identification mentioned above also adopted the same mutual audit process. For confirmability, rationales for identifying and adopting the codes and themes, as well as finalizing the use of specific quotes for illustrations of the thematic results were revealed and shared among the team members to seek objective consensus. Therefore, it took a long time for the findings to be repeatedly reviewed by team members. For instance, to achieve “audit trails”, the team kept records of the audio and transcript data from the interviews, as well as field notes, and asked the team members involved in coding to keep and share reflexive comments in discussions related to coding and data analysis. The reflexive ideas were also shared and discussed among other team members when finalizing the findings as presented in the paper.

## 4. Results

Participants in Shenzhen and Hong Kong had similar views on conceptualizing healthy ageing. However, when talking about the influences of retirement on their healthy ageing, the different cultural contexts and welfare systems in Shenzhen and Hong Kong constructed different experiences of retirees in certain domains of healthy ageing. The key results on the major themes that emerged are summarized in Table 2.

### 4.1. Domains of Healthy Ageing

Participants in Shenzhen (n = 12) and Hong Kong (n = 13) frequently mentioned mental and physical capacity as two main domains of healthy ageing. Some participants in Shenzhen (n = 4) and most (n = 11) in Hong Kong emphasized the interplay between mental and physical states and believed that psychological conditions impacted on one’s physical and functional abilities. A participant said: “*I truly value mental health since a positive emotion can benefit physical health naturally… As long as I am in good mood, my body will be well*” (male, 56 years old). Chinese older adults’ emphasis on mental health indicated cognitive advancement with increased understanding of and determination for healthy ageing. 

### 4.2. Purposes of Promoting Healthy Ageing

Six participants in Shenzhen and eleven participants in Hong Kong further elaborated on the purposes of promoting healthy ageing. They indicated that promoting healthy ageing was for maintaining independent living and avoiding reliance on family members. A participant stated: “*I regard healthy ageing as the self-care capacity because I can’t rely on my families to provide care when I am sick, lying in the bed for a long term*” (female, 62 years old). Thus, in addition to highlighting quality of life, participants also expressed that the purpose of healthy ageing was to reduce the burden on family members.

### 4.3. Effects of Retirement on Healthy Ageing 

Emotional well-being—Nearly half of retirees in Hong Kong (n = 6) and Shenzhen (n = 6) gained a sense of relief when they entered retirement. Escaping from heavy workloads and tight schedules, retirees had more free time to develop their own hobbies and interests. As one said: “*My work pressure was really heavy which requires careful consideration and endless communication … But the tension disappeared after retirement, and I feel released*” (male, 62 years old). Another respondent explained: “*I once spent a hard time in making living when I was young… But now I could engage in something that I am really interested in, and I really enjoy my retirement life*” (female, 56 years old). However, for some participants in Shenzhen (n = 2) and Hong Kong (n = 4), a sudden loss of social identity and increased leisure time made them feel anxious and upset. As one explained: “*After I retired, I felt very bored … It is just like we were abandoned and useless*” (female, 58 years old). Another respondent also expressed the same view: “*I lost my job, and I can’t see my colleagues every day, so I had a sense of loss, which made me wonder why my company gave up us so early*” (male, 68 years old).

Physical health—Quite a few of participants in Shenzhen (n = 6) and Hong Kong (n = 8) believed that their physical health would decline after retirement while their awareness of promoting health had increased. Retirees reported that they began to proactively pay more attention to personal lifestyle, diet, physical activities, and health information. One participant said: “*I used to drink a lot, but after retirement, I recognized that my physical health declined, so I drink less. I also watch TV programs about health and listen to health talks now*” (male, 66 years old). Similarly, another participant pointed out: “*After retirement, I was worried about physical deterioration and acute disease, so I started to focus on maintaining my physical health and increasing social participation”* (female, 58 years old). Participants took various measures such as regular medical examinations, balanced diets, and physical exercise to maintain their physical health.

Social network—Participants reported that the size of their social network had shrunk since retirement, but the quality and depth had advanced. Some of participants in both cities (Shenzhen: n = 4, Hong Kong: n = 4) mentioned that relationships with previous colleagues were directly influenced by retirement. Friendships among colleagues were expected to weaken or dissolve as older people left workplaces. As one indicated: “*I had less contact with my colleagues after retirement. But I also knew some new friends through participating in community activities*” (male, 67 years old). As a result of a shrinking social circle and having more time for social participation after retirement, some participants (Shenzhen: n = 4, Hong Kong: n = 4) felt the need to nurture relationships with people who shared similar interests outside the workplace. 

In contrast to alienation from previous colleagues, some retirees in both cities (Shenzhen: n = 5, Hong Kong: n = 8) increased the frequency of catching up with friends and families. Increased interactions with family members (including spouses, adult children, parents, etc.) strengthened older adults’ family relationships. A participant explained: “*I used to feel exhausted at work and had no time to communicate with my mother. But now I enjoyed a closer relationship with my mother through more communication*” (female, 54 years old). Furthermore, expectations on emotional supports from adult children were frequently mentioned by retirees in Shenzhen (n = 6), while participants in Hong Kong did not express such expectations. Compared to relatively independent relationships with children for retirees in Hong Kong, older people in Shenzhen had a greater demand for emotional support from adult children. A participant said: “*My daughter is a filial child because she will buy good things for me, but I prefer to receive love and care from my daughter … I am gradually stepping into old age … I hope my daughter could realize that and express care to me*” (female, 57 years old). 

Financial security—Some of retirees in Shenzhen (n = 5) and Hong Kong (n = 5) experienced financial stress because they were not entitled to certain social welfare provisions. Retirees in Hong Kong expressed that they had less income after retirement; however, their expenses for medical service and transportation increased. It was even more stressful for people who entered early retirement before 65 years old, as they were not eligible to medical benefits and transportation subsidies provided to senior citizens. A participant said: *“There is a gap in the social welfare provided for retirees, I am just 59 years old, and I am not allowed to enjoy welfare until 65 years old. However, I like going out, but transportation fee is a burden to me because I do not have income. Also, without healthcare insurance offered by my employer, I do not dare to see a doctor”* (female, 58 years old). In Shenzhen, financial stress was only mentioned by migrated retirees, as the amounts of pension varied among cities; therefore the pensions gained in their hometowns could barely sustain their lives in Shenzhen, as one stated: “*My wife and I were retired in the hometown but the expenses in Shenzhen are high, so our pensions are just enough for our living here*” (male, 56 years old). Worse still, without household registration, migrated retirees could not enjoy social welfare until 65 years old, as one stated: “*the eligible age for migrants to enjoy social benefits is higher than local people, which is unfair to us*” (female, 50 years old). 

Social participation—Driven by the motivation to reduce financial pressure and pursue a meaningful later life, participants in Hong Kong tended to have a strong desire for social activities (n = 7) and re-employment (n = 9). There are a series of benefits that retirees can draw from re-entering the labor market: supporting livelihood, keeping up with the social rhythm, and expanding social circles after retirement. One respondent said, “*A job is quite important for older adults because if we are unemployed, the motivation and energy of life will decline*” (n = male, 58 years old). Therefore, retirees expected to be provided with re-employment opportunities. However, age discrimination hindered them from entering the labor market again. One participant explained: “*I used to apply for a job, and I think I am qualified … The employer firstly asked me how old I was and reminded me that I had to consider my age … I am extremely disappointed with Hong Kong, a society with age discrimination*” (female, 62 years old). On the contrary, most of the retirees (n = 7) in Shenzhen had actively participated in community activities. Even though the remaining five participants did not have social participations due to the responsibility of caregiving for grandchildren/parents, all retirees in Shenzhen (n = 12) had expressed interests in community activities rather than re-employment. Participation in community activities can improve the physical and mental health of older people as well as extending their social networks. Therefore, participation in community activities was recommended as a way for promoting healthy ageing. As one said: “*Community activities are beneficial for physical and mental health of older people. Also, participation in these activities motivates older people to go out and extend their social networks. In this case, they can find someone to share their feelings*” (male, 67 years old). 

## 5. Discussion

The findings show that recent retirees in Hong Kong and Shenzhen perceived maintaining an independent life and reducing burden on family members as the main purposes of healthy ageing. In other words, retirees in both cities acknowledged that the domains of healthy ageing included physical, mental, social health, and financial security. Retirement produced similar effects on physical and mental health for retirees in both cities. Specifically, older people experienced a decline in physical health but raised health awareness after retirement. For mental health, the effects of both identity loss and stress relief caused by retirement were documented in this study. Due to differing retirement systems in Shenzhen and Hong Kong, the current study found that older people in Hong Kong reported higher stress levels of financial security after retirement, which further motivated them to actively engage in social activities, particularly in the re-employment labor market. 

Shaped by the same traditional cultural background, Chinese retirees in both cities shared common perspectives toward understanding healthy ageing. Functional aspects such as taking care of themselves and being independent are highly valued by retirees, which is identical to the WHO’s definition of maintaining functional ability [1]. The findings show that, for Chinese people, the primary concern is to maintain functional ability not just for the sake of personal benefits but also for the need to avoid becoming a burden to family. Chinese people are embedded with a deep sense of familism and collectivism; they generally have a wide range of connections with family, groups, and society, and have developed various social relationships. The current cohort of retirees belongs to a generation that is deeply influenced by the ideas of familism and collectivism, which can largely explain why their views toward retirement and healthy ageing were so much connected with the interests of the social groups that the retirees were affiliated with. 

This study finds that retirement has similar influences on physical and mental health under voluntary and mandatory retirement systems in Hong Kong and Shenzhen, respectively. For physical health, recent retirees tended to report a poorer health status compared to that of their working period, which was consistent with a previous study [13]. Simultaneously, in view of the declined physical health, retirees raised their health awareness and adopted healthier behaviors (e.g., personal lifestyle, diet, physical activities, and health literacy). However, for mental health, this study observed that older adults experienced the co-existence of a sense of relief and identity loss after retirement. This finding echoed the mixed results in previous studies that retirement is associated with both stress relief and identity disruption in older people [15,16,17].

With respect to social networks, retirees reported that the size of their social connection had shrunk, but the quality and depth had advanced. This result could be supported by a previous finding that retirement transition decreased the number of peripheral or weak relationships (colleagues or friends), while intensifying strong social ties (family members or intimate friends) [39]. In addition, the importance of intergenerational relationships was also discussed by retirees. Influenced by the traditional Chinese notion of filial piety, retirees in Shenzhen emphasized more the influence of intergenerational relationships on their healthy ageing than retirees in Hong Kong. Owing to the integration of western and Chinese cultures, filial piety has shifted from parental authority to equality in intergenerational relationships in Hong Kong [40]. Therefore, retirees in Hong Kong may have fewer expectations regarding support provision from adult children. However, the findings show that expectations of filial piety in Mainland China also changed as retirees talked about the need to maintain independence and put more emphasis on emotional support from adult children. Changing expectations of filial piety may result from changes in family structures. Particularly, the One-Child policy has rendered caring for older parents burdensome for the only child [41]. In this study, the majority of participants in Shenzhen had only one child. Therefore, retirees in Shenzhen would perceive healthy ageing as key to reducing the burden on their children, and they expected only emotional support from their children. 

The influence of different social welfare systems in Hong Kong and Shenzhen was reflected in the effect of retirement on financial security. For instance, older adults in Hong Kong reported more stress and worries about financial burdens after retirement, which is primarily attributed to the lack of a universal pension scheme. Although the MPF system has been well established in Hong Kong, it has been criticized that MPF was only eligible for working people and not sufficient to cover retirement life [42]. In addition, there was a welfare gap for people who experienced early retirement (below 65 years old) under the current social welfare system in which old-age subsidies (e.g., Old Age Allowance, Senior Citizen Card, and Health Care Voucher) were only available for people aged 65 years old or above. Previous research confirmed the financial concerns of retirees in Hong Kong, indicating that older people had substantial fear and insecurity about future finances, and a strong desire to seek re-employment opportunities in the labor market [43]. 

On the contrary, covered by the state-led pension insurance and medical insurance [30], the financial burden of older people in Shenzhen was greatly released after retirement. It shows that the Urban Employee Pension is sufficient to cover the living expenses of most beneficiaries [44]. Hence, retirees in Shenzhen tended to enjoy recreational or leisure activities. However, as Shenzhen is a migrant city, the household registration system may limit migrated retirees’ access to social welfare and sufficient pension. Along with the ongoing urbanization process in mainland China, increasing internal migration and social mobility has introduced more older migrants to urban cities [45]. Previous studies also identified pension inequality and local–migrant gaps between older migrants and local residents in the accessibility to welfare (e.g., healthcare utilization) in Mainland China [46,47]. 

The results of this study informed the following implications to better promote healthy ageing after retirement. First, older adults need to gain a better understanding of the role and impacts of retirement in their life course so as to learn and develop strategies for living through their retirement life stage. Programs on retirement planning should be further developed by service providers, thus helping older people to make advanced planning and adapt to retirement life emotionally and physically. Second, given the financial hardship of retirees in Hong Kong, it is suggested that policymakers should reinforce a multi-pillar retirement-protection system and establish supportive employment schemes that facilitate old-age labor participation. In addition, special attention should be paid to addressing the gap between the retirement age and the eligible allowance age in Hong Kong. Third, to narrow the welfare gap between older migrants and local residents in mainland China, more efforts should be made to integrate public welfare benefits regardless of *Hukou* status. In addition, more community-based recreational activities are supposed to be organized to enhance neighborhood cohesion and social inclusion of older migrants. 

Several limitations have been noted in this study. First, while the results show the different perspectives of migrated retirees and local people on retirement and healthy ageing, future studies should include more migrated retirees to explore their viewpoints. Second, this study examined other adults’ perception of healthy ageing by recruiting a small group of participants with different socio-demographic characteristics; future studies could be conducted in a more in-depth way by recruiting a larger number of participants with similar demographic and social backgrounds. Third, relevant quantitative studies could be designed to provide more empirical support for the results regarding the impacts of retirement on various domains of healthy ageing. 

## 6. Conclusions

This study revealed older people’s perception of healthy ageing and how the socio-cultural context influences the effect of retirement on healthy ageing. From the perspectives of retirees, the concept of healthy ageing covered physical, mental, social, and financial domains. Generally, retirement declined physical health (in parallel with raised awareness of health promotion), reduced mental distress, triggered anxiety of identity loss, and shrank peripheral social networks of retirees. In addition, social welfare systems in Shenzhen and Hong Kong have different impacts on financial security and social participation. Older people in Hong Kong reported higher stresses of financial security after retirement. Correspondingly, they were more likely to engage in social activities, particularly in the re-employment labor market. For retirees in Shenzhen, the welfare gap between local residents and older migrants was highlighted. Overall, based on the holistic domain of healthy ageing, this study provided a rich picture of retirement effects on the physical, mental, social, and financial well-being of older adults in Shenzhen and Hong Kong. The results of this study suggested that socio-cultural context should be taken into consideration in examining the effect of retirement on healthy ageing. 

## Figures and Tables

**Figure 1 ijerph-20-02820-f001:**
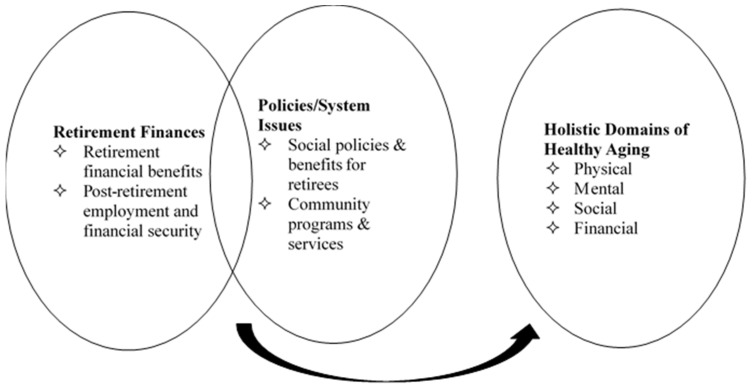
Conceptual framework of current study.

**Table 1 ijerph-20-02820-t001:** Demographic profile of participants (n = 25).

Characteristics of Participants	Shenzhen n = 12	Hong Kong n = 13
Age		
50–55 years old	4	2
56–60 years old	4	5
61–65 years old	4	6
Gender		
Male	5	6
Female	7	7
Educational level		
Middle school and below	7	3
College and above	5	10
Marital status		
Married	12	11
Single	0	1
Divorced	0	1
Number of children		
None	0	2
One	7	3
Two and above	5	8
Household register		
Local	6	13
Migrant	6	0
Occupation before retirement		
General employee	3	7
Administrative-level manager	1	1
Technician	3	1
Civil servant	1	2
Military personnel	1	0
Social worker	1	1
Factory worker	1	0
Freelance worker	1	0
Unemployed	0	1

**Table 2 ijerph-20-02820-t002:** Summaries of results.

Themes	Similarities in Sub-Themes	Differences in Sub-Themes	
	Shenzhen and Hong Kong	Shenzhen	Hong Kong
Domains of healthy ageing	Physical health		
	Mental health		
Purpose of healthy ageing	To maintain independent living		
	To reducing burden on family members		
Influences of retirement on healthy ageing			
Emotional well-being	Sense of relief		
	Sense of loss		
Physical health	Deterioration of health		
	More health-promotion behaviors		
Social network	Shrinkage of networks with colleagues		
	Increased contact with friends		
	Increased interactions with family	Expectations of emotional support from children	Little mention of expectations from children
Financial security		Increased financial stresses mentioned by migrants because they possess lower levels of pension and have restrictions on enjoying welfare	Increased financial stresses because people with early retirement are not entitled to enjoy healthcare and transport subsidies
Social participation	Increased social participation	Desire for community activities to maintain health	Desire for re-employment to reduce financial stresses

## Data Availability

Data sharing is unavailable due to privacy reason.

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
