# Peer review of "Experience of Chinese Recent Retirees on the Effects of Retirement on Healthy Ageing in Shenzhen and Hong Kong"

_ijerph, 2023, doi:10.3390/ijerph20042820_

Round 1
Reviewer 1 Report
Thank you for the possibility to read this manuscript. The aim of this study was to examine how the retirement of older Chinese in different socio-cultural contexts differs in terms of its implications on their health as they age after retirement. The study showed how the socio-cultural context influences people’s conception of healthy ageing and the effect of retirement of healthy ageing. The manuscript is mainly well-written but need some revisions and I hope the questions and comments below will be helpful.
TITLE: This is a qualitative study about people’s experiences of the effects of retirement on healthy ageing of recent retirees in …….more than the effects…..Please revise the title.
On the first page in the Background section, second paragraph you write both cities on the third line but in the same sentence you write mainland China and Hong Kong and that are not two cities.
Please describe how you have used the conceptual framework you present.
How were the first participants selected? The following were selected through snowball sampling.
How many people refused to participate or dropped out (and what was the reason)?
How were participants approached?
Were the interview questions pre-tested?
You need to demonstrate that data analysis has been conducted in a precise, consistent, and exhaustive manner through disclosing the methods of analysis with enough detail, please revise this part.
How did you ensure trustworthiness? One way is that the research process is clearly documented for example through an audit trail but please describe other ways too.
I also think it is a good idea to summarize the themes and categories in a table or figure in the beginning of the results. This hopefully makes it clearer what you describe in the result section - themes or categories.
According to the author instructions limitations of the work should be highlighted in the discussion not in the conclusions. Please move this part and develop it further.
Reviewer 2 Report
Summary
This study documents healthy aging in Shenzhen and Hong Kong using narrative interviews. The authors interviewed 25 senior people that are comparable in the two cities and summarized the transcripts to do qualitative analysis. The analysis covers the physical, mental, social, and financial domains of healthy aging. The results show that functional ability and in-family relationships are important components in the context of healthy aging, which adds new Chinese facts to the literature.
Major issues
1. This paper did many fieldworks and obtained valuable narrative data. Compared with ordinary statistics, such data provide a more comprehensive view of the situation of healthy aging in the region of the Pearl River Delta. But one might be interested in some quantitative results accompanying the current qualitative conclusions to build more intuitions. Descriptive statistics about income, consumption, and household wealth are strongly suggested if they are available.
2. This paper studies the impact of retirement on healthy aging in China while using Shenzhen and Hong Kong as two representative cities. But the cited works in the literature review are mostly about other countries. One may be interested in the development of this literature in the Chinese context and how this paper contributes to the literature. Is this paper a pioneer in documenting healthy aging in the Chinese context? Do the qualitative conclusions of this paper differ from similar studies? The authors are supposed to address this issue carefully.
3. To properly represent the overall situation of healthy aging in China, the interviewees are supposed to be representative enough. Shenzhen and Hong Kong are two highest-income cities. The authors are supposed to address why the two cities are representative. For example, this paper samples senior people from Shenzhen with permanent residences while repeating the immigrants in Shenzhen. One may be interested in the sampling bias because people with permanent residences in Shenzhen are likely to have higher income levels than others. Would such bias affect the analysis, and how?
4. Section 5 cites many original transcripts from the interviewees without giving a big picture of the data. One may be very interested in how the data really look like, the distribution of question responses, but not simply pieces of records. The authors are strongly recommended to add a section about the big picture of re-organized and aligned data.
5. Section 5 puts some claims without data support. E.g. “On the contrary, most retirees in Shenzhen expressed more interest in community activities, rather than re-employment. Participating in community activities could delight older people’s bodies and minds and enable them to meet new friends” The authors are suggested to provide more data evidence to support such claims..
Other issues
1. The English need to be somewhat improved. Please correct all typos (not fully listed in this report) carefully.
2. Section 2.1 says that “However, the majority of previous studies investigate the influence of retirement on health under the voluntary retirement system. China mainly implements a mandatory retirement system, which may trigger different influences of retirement on health”. How this difference affects this paper’s results? The authors should more address this issue in discussion or show the robustness of the conclusions against this issue.
3. Section 4.1 claims two research questions. This paper is supposed to include the first question into contribution if it is the first study that documents Chinese facts.
4. Section 4.2 says, “Participants were the active character to lead the content and peace of interviews.” Here the word peace is supposed to be pace.
5. Table 1 is ill-formatted. Please align variable names, and value labels. Please be consistent with “n=” or “N=”.
Round 2
Reviewer 1 Report
Thank you for your revision. You have revised everything I asked for but I still think that the steps of the analysis could have been clearer, and how you have strengthen credibility, dependability, confirmability and transferability could be described in more detail.
Author Response
Thanks for the comment, more details of the data analysis process were added in Data Analysis part (p.6). Also, we have provided more illustrations on how to achieve crediability, transferability, dependability and confirmability, in p.6.
Reviewer 2 Report
I have no another question.
Author Response
Thanks for the review and comments. Spell has been checked and corrected. No other revision was made based on reviewer's comment.